# Spatial distribution and determinants of the change in pre-lacteal feeding practice over time in Ethiopia: A spatial and multivariate decomposition analysis

Achamyeleh Birhanu Teshale[1]*, Misganaw Gebrie Worku[2], Getayeneh Antehunegn Tesema[1]

1 Department of Epidemiology and Biostatistics, Institute of Public Health, College of Medicine and Health Sciences, University of Gondar, Gondar, Ethiopia, 2 Department of Human Anatomy, University of Gondar, College of Medicine and Health Science, School of Medicine, Gondar, Ethiopia

* achambir08@gmail.com

## Abstract

### Background

Pre-lacteal feeding persists in low and middle-income countries as deep-rooted nutritional malpractice. It imposes significant negative consequences on neonatal health, including increased risk of illness and mortality. Different studies revealed that pre-lacteal feeding practice is decreased over time. Even though different studies are done on the prevalence and determinants of pre-lacteal feeding practice, up to our knowledge, the spatial distribution and the determinants of the change in pre-lacteal feeding practice over time are not researched.

### Objective

To assess the spatial distribution and determinants of the change in pre-lacteal feeding practice over time in Ethiopia.

### Methods

We used the Ethiopian demographic and health surveys (EDHSs) data. For this study, a total weighted sample of 14672 (5789 from EDHS 2005, 4510 from EDHS 2011, and 4373 from EDHS 2016) reproductive-age women who gave birth within two years preceding the respective surveys and whoever breastfeed were used. The logit-based multivariate decomposition analysis was used to identify factors that contributed to the decrease in pre-lacteal feeding practice over the last 10 years (from 2005 to 2016). Using the 2016 EDHS data, we also conducted a spatial analysis by using ArcGIS version 10.3 and SaTScan version 9.6 software to explore the spatial distribution and hotspot clusters of pre-lacteal feeding practice.

**Data Availability Statement:** It is ethically not acceptable to share the DHS data sets to third parties. However, anyone who want the data set can access from the Measure DHS program at www.dhsprogram.com, through legal requesting.

The authors had no special access privileges others would not have.

**Funding:** The author(s) received no specific funding for this work.

**Competing interests:** The authors have declared that no competing interests exist.

**Abbreviations:** ANC, Antenatal Care; C, coefficient; E, Endowment; EAs, Enumeration areas; EDHS, Ethiopian Demographic and Health Survey; EDHS, Ethiopian Demographic and Health Surveys; LLR, Log likelihood Ratio; NNP, National Nutrition Program; RR, Relative Risk; WHO, World Health Organization.

## Result

Pre-lacteal feeding practice was decreased from 29% [95% Confidence interval (CI): 27.63–29.96%] in 2005 to 8% [95% CI: 7.72–8.83%] in 2016 with annual rate of reduction of 7.2%. The overall decomposition analysis showed that about 20.31% of the overall decrease in pre-lacteal feeding practice over the last 10 years was attributable to the difference in composition of women (endowment) across the surveys, while, the remaining 79.39% of the overall decrease was explained by the difference in the effect of characteristics (coefficient) across the surveys. In the endowment component, the difference in composition of residence, perception of distance from the health facility, maternal educational level, wealth status, occupation, ANC visit, place of delivery, the timing of breastfeeding initiation, and wanted last-child/pregnancy were found to be significant contributing factors for the decrease in pre-lacteal feeding practice. After controlling for the role of compositional changes, the difference in the effect of distance from the health facility, wealth status, occupation, antenatal care (ANC) visit, and wanted last-child/pregnancy across the surveys were significantly contributed to the observed decrease in pre-lacteal feeding practice. Regarding the spatial distribution, pre-lacteal feeding practice was non-random in Ethiopia in which the primary and secondary clusters' of pre-lacteal feeding identified in Somalia and the Afar region respectively.

## Conclusion

Pre-lacteal feeding practice has shown a significant decline over the 10-year period. Program interventions considering women with poor maternal health service utilization such as ANC visits, women with poor socioeconomic status, women with an unintended pregnancy, and women from remote areas especially at border areas such as Somali and Afar could decrease pre-lacteal feeding practice in Ethiopia.

## Background

The World Health Organization (WHO) and the National Nutrition Program (NNP) of Ethiopia have recommended starting breastfeeding within one hour of birth, breastfeeding exclusively for the first six months of life, and continuing breastfeeding up to two years of age [1–3].

Pre-lacteal feeding, however, is a barrier to implementing exclusive breastfeeding practices and initiating breastfeeding promptly [4–8]. Prelacteal foods are foods (can be water only, water-based such as rice water, herbal mixture, and milk-based such as animal milk and infant formula) given to the newborn baby, often during the first three days of life, before breastfeeding is developed or started [1,9,10].

Pre-lacteal feeding continues in developing countries as deep-rooted nutritional malpractice and results in negative neonatal health outcomes, including increased risk of illness and mortality (23–25). It decreases the immunological benefits of colostrum provided in the first three days after delivery, thus raising the susceptibility of the newborn to infection (26). Also, by exposing infants to infected foods, utensils, water, or hands, pre-lacteal feeding may be a direct cause of illness (23). By interfering with the priming of the gastrointestinal tract, intellectual and physical growth, as well as by reducing the immune status, pre-lacteal feeding can also affect neonatal health (23, 24). In addition, pre-lacteal feeding interrupts mother-infant bonding and reduces a mother's near skin-to-skin contact with her son (25, 26).

Pre-lacteal feeding is still a major public health problem. Worldwide the prevalence of pre-lacteal feeding ranges from 12.3% in Timor-Leste to 85.2% in Nigeria [6,11–15]. In Ethiopia, pre-lacteal feeding practice is also a devastating problem that ranges from 14.2% in the Mettu district to 38.8% in Raya Kobo [16–20].

Evidences revealed that maternal age, maternal educational level, socioeconomic status, exposure to media, antenatal care (ANC) visit, place of delivery, cesarean delivery, the timing of breastfeeding initiation, parity, sex of the child, distance from the health facility, and residence are among the different factors contributing for pre-lacteal feeding practice [15,17,18,21–26].

Different studies also revealed that pre-lacteal feeding practice is decreased over time. In rural Bangladesh, it decreased from 88.0% in 2004 to 24.7% in 2019 [23]. Another study in Nigeria also revealed that pre-lacteal feeding practice is decreased from 66% in 2003 to 55% in 2013 [22]. In Ethiopia, pre-lacteal feeding practice has shown a significant decrease from 29% in 2005 to 8% in 2016 [27,28].

While different studies are carried out on the prevalence and determinants of pre-lacteal feeding practice, the spatial distribution, and the contributing factors for the drastic changes in the practice of pre-lacteal feeding in Ethiopia are not researched. Therefore, we aimed to assess the spatial distribution and determinants of the change in pre-lacteal feeding practice in Ethiopia. The findings of this study can be used as an input for policymakers to plan strategies and intervene in this devastating public health problem.

## Methods

### Data source, sampling procedure, and study population

We used the three Ethiopian demographic and health surveys (EDHSs) (2005, 2011, and 2016) data, which are the nationally representative surveys performed in Ethiopia. In each of the surveys, a two-stage cluster sampling was employed. In the first stage, 540 Enumeration Areas (EAs) for EDHS 2005, 624 EAs for EDHS 2011, and 645 EAs for EDHS 2016 were randomly selected proportional to the EA size and, on average, 27 to 32 households per EAs were selected in the second stage. A total weighted sample of 14672 (5789 from EDHS 2005, 4510 from EDHS 2011, and 4373 from EDHS 2016) reproductive-age women who gave birth within two years preceding the respective surveys and whoever breastfeed were used for this study. There is detailed and comprehensive information relating to the sampling process and other information in each survey report [27–29].

### Variables of the study

The outcome variable was feeding of the child other than breast milk within three days, which was a binary outcome variable coded as "1" if the mother gave anything other than breast milk and "0" if a mother gave nothing for her newborn child within three days.

The independent variables included (after searching of literatures) for our study were region, place of residence, perception of distance from the health facility, age, educational level, wealth index, occupation, mass media exposure, parity, ANC visit, place of delivery, delivery by cesarean section, size of the child at birth, and timing of initiation of breastfeeding.

**Operational definitions.**    *Mass media exposure*: Created by combining whether a respondent reads a newspaper, listen to the radio, and watch television and coded as yes (if a woman had exposed to at least one of these media) and no (if women were not exposed to at least one of the media).

*Size of the child at birth*: It is defined as the size of the child during delivery, which is based on the mere report of mothers and categorized in the surveys as very small, small, average,

large, and very large and recoded as average, small (includes very small and small), and large (includes large and very large) for this analysis.

The other independent variable definitions are self-explanatory and more information about these variables can get from the EDHS 2016 report [28].

## Data management and statistical analysis

The data were extracted and recoded using Stata version 14. Throughout the analysis, the data were weighted to make it representative and to provide better statistical estimates.

**Trend and decomposition analysis.** The trend and multivariate decomposition analyses were done using Stata version 14. The trend of pre-lacteal feeding practice was examined separately for the periods 2005–2011, 2011–2016, and 2005–2016. The trend of pre-lacteal feeding in each of the selected sociodemographic characteristics of respondents was also analyzed using descriptive analysis.

The multivariate decomposition analysis technique was used to analyze the difference in pre-lacteal feeding practice between two points in time (2005 and 2016). It is widely practiced in public health studies to identify components of a change over time and identify contributing factors for the change [30,31]. The analysis decomposes the differences in pre-lacteal feeding practice over time into two components (the endowment part and coefficient part).

For our study, the 2016 EDHS data was appended to the 2005 EDHS data using the "append" Stata command, and the logit based multivariate decomposition analysis (using mvdcmp STATA command) was used to identify factors that contributed to the decrease in pre-lacteal feeding practice over the last 10 years. Therefore, the observed decrease in pre-lacteal feeding practice was additively decomposed into differences due to endowment/characteristic and differences due to coefficient/effects of the characteristic component.

In doing the decomposition analysis, the Logit or log-odd of pre-lacteal feeding practice is taken as [31]:

$$\text{Logit}\ (2005) - \text{Logit}\ (2016) = F\ (X_{2005}\ \beta_{2005}) - F\ (X_{2016}\ \beta_{2016})$$

$$= \underbrace{\{F\ (X_{2005}\ \beta_{2005} - F\ X_{2016}\ \beta_{2005})}_{E} + \underbrace{\{F\ (X_{2016}\ \beta_{2005}) - F\ (X_{2016}\ \beta_{2016})\}}_{C}$$

In which, the "E" component is the part of the differential due to differences in characteristics while the "C" component refers to the part of the differential attributable due to differences in coefficients or effects of characteristics.

**Spatial analysis.** We conducted a spatial analysis using ArcGIS version 10.3 and SaTScan version 9.6 software. To assess whether the spatial distribution of pre-lacteal feeding practice was random or non-random (spatial autocorrelation), Global Moran's I statistic was used.

Kriging spatial interpolation technique was used to predict pre-lacteal feeding practice in the un-sampled areas based on the values from sampled measurements. Besides, Getis Ord Gi* statistical hotspot analysis was done to identify the significant hot spot areas (areas with high rates of pre-lacteal feeding practice) and cold spot areas (areas with lower rates of pre-lacteal feeding practice).

Moreover, we used Bernoulli based spatial scan statistical analysis to detect statistically significant clusters. To fit the model women who gave anything within three days for the newborn were taken as cases and those who gave nothing were taken as controls. The primary and secondary clusters were identified and p values were assigned and ranked using their log-likelihood ratio (LLR) test based on the 999 Monte Carlo replications. Areas with high LLR and significant p-value were considered as clusters with higher rates of pre-lacteal feeding practice

and the spatial window with the highest significant LLR test statistic was defined as the most likely (primary) cluster.

### Ethical considerations

Since this is a secondary analysis of the Demographic and Health Survey (DHS) data, ethical approval was not necessary. However, we registered and requested the datasets from DHS on-line archive and received permission to access and download the data files. Moreover, for Geographic information system coordinates, the coordinates are only for the enumeration area (EA) as a whole and the measured coordinates were randomly displaced within a large geographic area so that no particular enumeration areas can be identified.

## Results

### Socio-demographic characteristics of respondents

In all the three consecutive EDHSs, the majority of the study participants were in the age group 25 to 34 years. With regard to residence, there was a slight increment of urban residents (from 8.13% in 2005 to 11.85% in 2016). About 25.67% of women in 2005 did not perceive distance from the health facility as a big problem and this figure rose to 39.96% in 2016. Regarding educational status, in the first two surveys about three-quarters and two-thirds (78.59% in 2005 and 66.79% in 2011) respectively were not educated, while 60.36% were not educated in EDHS 2016. The percentage of working women has been increased from 29.24% in 2005 to 41.70% in 2016. In addition, the timely initiation of breastfeeding increases from 73.63% in 2005 to 84.10% in 2016. Generally, the proportion of women with all explanatory variables except wealth index, region, parity, size of children, and sex of child significantly varies across the surveys (2005–2016) (**Table 1**).

### Overall trends in pre-lacteal feeding practice in Ethiopia, 2005–2016

Over the 10-year period, pre-lacteal feeding practice has shown a substantial decrease from 29% [95% Confidence Interval (CI): 27.63–29.96%] in 2005 to 8% [95% CI: 7.72–8.83%] in 2016 with the annual rate of reduction of 7.2% (**Fig 1**).

### Trends of pre-lacteal feeding by selected characteristics of respondents

The trends in pre-lacteal feeding practice showed variation according to the respondent's characteristics. A decline in pre-lacteal feeding practice was observed in women with all of the categories of variables. Over the past 10-years, pre-lacteal feeding practice has decreased significantly in all regions of Ethiopia, except in the Somalia region, where the proportion of pre-lacteal feeding practice has increased by 18.09% (**Table 2**).

### Decomposition analysis

The overall decomposition revealed that about 20.31% of the overall decrease in pre-lacteal feeding practice over the 10-year period was attributable to the difference in characteristics (composition) of women across the surveys with the remaining 79.69% attributable to the difference in the effect of characteristics (coefficient) across the surveys (**Table 3**). In the endowment component, the difference in composition of women with respect to residence, perception of distance from the health facility, educational level, wealth status, occupation, ANC visit, place of delivery, timing of breastfeeding initiation, and wanted last-child/pregnancy across the surveys were significant contributing factors for the decrease in pre-lacteal feeding practice (**Table 4**).

**Table 1. Percentage distribution of socio-demographic characteristics of respondents 2005, 2011, and 2016 Ethiopia Demographic and Health Surveys.**

| Characteristics | 2005 [N = 5789] | 2011 [N = 4510] | 2016 [N = 4373] |
|---|---|---|---|
| Residence | | | |
| Urban | 8.13 | 13.56 | 11.85 |
| Rural | 91.87 | 86.44 | 88.15 |
| Distance from the health facility | | | |
| Big problem | 74.33 | 74.29 | 60.04 |
| Not a big problem | 25.67 | 25.74 | 39.96 |
| Age (years) | | | |
| 15–24 | 27.86 | 30.53 | 29.11 |
| 25–34 | 46.69 | 49.13 | 50.74 |
| 35–49 | 25.44 | 20.34 | 20.15 |
| Educational level | | | |
| No education | 78.59 | 66.79 | 60.36 |
| Primary | 16.5 | 28.52 | 30.94 |
| Secondary & above | 4.91 | 4.7 | 8.7 |
| Wealth status | | | |
| Poor | 41.84 | 45.38 | 45.24 |
| Middle | 22.25 | 20.66 | 21.03 |
| Rich | 35.92 | 33.96 | 33.74 |
| Occupation | | | |
| Not working | 70.76 | 49.09 | 58.3 |
| Working | 29.24 | 50.91 | 41.7 |
| Media exposure | | | |
| No | 62.64 | 40.67 | 65.34 |
| Yes | 37.36 | 59.33 | 34.66 |
| Parity | | | |
| Primiparous | 16.89 | 18.57 | 20.8 |
| Multiparous | 42.17 | 44.73 | 42.04 |
| Grand multiparous | 40.94 | 37.12 | 37.15 |
| ANC visit | | | |
| No | 71.63 | 50.51 | 35.18 |
| 01-Feb | 9,54 | 13.22 | 13.37 |
| 3 | 6.34 | 13.08 | 18.29 |
| 4 and above | 12.48 | 17.19 | 33.16 |
| Place of delivery | | | |
| Home | 93.64 | 88.54 | 64.42 |
| Health facility | 6.36 | 11.46 | 35.58 |
| Delivery by CS | | | |
| No | 98.85 | 98.14 | 97.47 |
| Yes | 1.15 | 1.86 | 2.55 |
| Size of the child at birth | | | |
| large | 29.3 | 29.72 | 29.46 |
| Average/normal | 41.72 | 38.33 | 41.93 |
| Small | 28.98 | 31.96 | 28.6 |
| Sex of the child | | | |
| Male | 51.43 | 52.31 | 48.77 |
| Female | 48.57 | 47.69 | 51.23 |
| Timing of BF initiation | | | |

(*Continued*)

**Table 1.** (Continued)

| Characteristics | 2005 [N = 5789] | 2011 [N = 4510] | 2016 [N = 4373] |
|---|---|---|---|
| Within one hour | 73.63 | 64.25 | 84.1 |
| After one hour | 26.35 | 35.75 | 15.9 |
| Wanted of the child | | | |
| Yes | 67.68 | 66.09 | 73.88 |
| No | 32.32 | 33.91 | 26.12 |
| Region | | | |
| Tigray | 6.45 | 6.06 | 7.19 |
| Afar | 0.93 | 0.93 | 1.01 |
| Amhara | 25.73 | 21.67 | 18.54 |
| Oromia | 37.23 | 43.4 | 44.42 |
| Somalia | 3.76 | 2.89 | 4.08 |
| Benishangul | 0.91 | 1.13 | 1.05 |
| SNNPR | 22.56 | 20.86 | 20.18 |
| Gambela | 0.3 | 0.38 | 0.25 |
| Harari | 0.21 | 0.25 | 0.24 |
| Addis Ababa | 1.58 | 2.09 | 2.61 |
| Dire Dawa | 0.33 | 0.33 | 0.42 |

Note: BF = Breastfeeding, SNNPR = Southern Nation Nationalities and People's Region.

An increase in the proportion of women living in urban area [β = 0.004617, 95% CI: 0.002943, 0.006291] and women who did not perceive distance from the health facility as a big problem [β = 0.005564, 95% CI: 0.002435, 0.008692] contributed a 3.26% and 3.93%, respectively for the change in pre-lacteal feeding practice. An increase in the composition of women from households with a middle wealth index over the survey period contributes to a significant change in pre-lacteal feeding practice [β = 0.003016, 95% CI: 0.001606, 0.004425]. A decrease

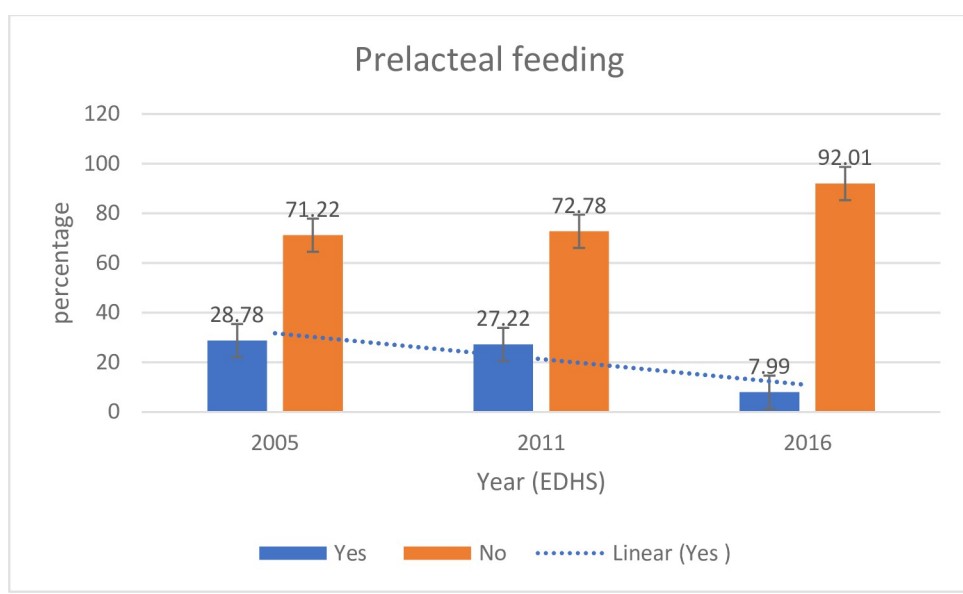

**Fig 1. Overall trends of pre-lacteal feeding in Ethiopia from 2005 to 2016.**

**Table 2. Trends in pre-lacteal feeding practice among reproductive-age women by selected characteristics, 2005, 2011, and 2016 Ethiopia Demographic and Health Surveys.**

| Characteristics | EDHS 2005 | EDHS 2011 | EDHS 2016 | Percentage point difference in practicing of prelacteal feeding | | |
|---|---|---|---|---|---|---|
| | | | | Phase I 2011–2005 | Phase II 2016–2011 | Overall 2016–2005 |
| Residence | | | | | | |
| Urban | 25.15 | 24.85 | 12.26 | -0.3 | -12.59 | -12.89 |
| Rural | 19.26 | 27.59 | 7.42 | 8.33 | -20.17 | -11.84 |
| Distance from the health facility | | | | | | |
| Big problem | 27.97 | 27.29 | 7.61 | -0.68 | -34.9 | -20.36 |
| Not a big problem | 31.11 | 27 | 8.56 | -4.11 | -18.44 | -22.55 |
| Educational level | | | | | | |
| No education | 28.84 | 29.98 | 8.41 | 1.14 | -21.57 | -20.43 |
| Primary | 26.67 | 21.16 | 6.66 | -5.51 | -14.5 | -20.01 |
| Secondary & above | 34.91 | 24.72 | 9.8 | -10.19 | -14.92 | -25.21 |
| Wealth status | | | | | | |
| Poor | 29.37 | 33.96 | 8.55 | 4.59 | -25.41 | -20.82 |
| Middle | 25.98 | 23.37 | 6.07 | -2.61 | -17.3 | -19.91 |
| Rich | 29.83 | 20.55 | 8.45 | -9.28 | -12.1 | -21.08 |
| Occupation | | | | | | |
| Not working | 27.71 | 26.05 | 7.94 | -1.66 | -18.11 | -19.77 |
| Working | 31.36 | 28.34 | 8.07 | -3.02 | -20.27 | -23.29 |
| Parity | | | | | | |
| Primiparous | 33.78 | 31.38 | 10.76 | -2.4 | -20.62 | -23.02 |
| Multiparous | 29.37 | 26.44 | 6.32 | -2.93 | -20.12 | -23.05 |
| Grand multiparous | 26.1 | 26.12 | 8.33 | 0.02 | -17.79 | -17.77 |
| ANC visit | | | | | | |
| No | 28.43 | 30.7 | 9.38 | 2.27 | -21.32 | -19.05 |
| 01-Feb | 30.22 | 28.35 | 10.37 | -1.87 | -17.98 | -19.85 |
| 3 | 26.72 | 23.17 | 5.74 | -3.55 | -17.43 | -20.98 |
| 4 and above | 30.73 | 17.96 | 6.8 | -12.77 | -11.16 | -23.93 |
| Place of delivery | | | | | | |
| Home | 29.21 | 28.03 | 8.46 | -1.18 | 19.57 | -20.75 |
| Health facility | 28.75 | 20.9 | 7.14 | -7.85 | -13.76 | -21.61 |
| Size of the child at birth | | | | | | |
| large | 29.61 | 24.03 | 7.4 | -5.58 | -16.63 | -22.21 |
| Average/normal | 27.55 | 24.29 | 7.67 | -3.29 | -16.62 | -19.88 |
| Small | 29.71 | 33.69 | 9.08 | 3.98 | -24.61 | -20.63 |
| Timing of BF initiation | | | | | | |
| Within one hour | 19.57 | 15.02 | 5.38 | -4.55 | -9.64 | -14.16 |
| After one hour | 54.51 | 49.14 | 21.79 | -5.37 | -5.37 | -32.72 |
| Wanted of the child | | | | | | |
| Yes | 29.13 | 28.26 | 8.05 | -0.87 | -20.21 | -21.08 |
| No | 28.2 | 25.18 | 7.82 | -3.02 | -17.36 | -20.38 |
| Region | | | | | | |
| Tigray | 29.59 | 25.84 | 6.27 | -3.75 | -19.57 | -23.32 |
| Afar | 39.84 | 30.7 | 40.08 | -9.14 | 9.38 | 0.24 |
| Amhara | 45.4 | 47.57 | 8.32 | 2.17 | -39.25 | -37.08 |
| Oromia | 24.8 | 22.18 | 4.16 | -2.62 | -18.02 | -20.64 |
| Somalia | 20.75 | 74.1 | 38.84 | 53.35 | -35.26 | 18.09 |

*(Continued)*

**Table 2.** (Continued)

| Characteristics | EDHS 2005 | EDHS 2011 | EDHS 2016 | Percentage point difference in practicing of prelacteal feeding | | |
|---|---|---|---|---|---|---|
| | | | | Phase I 2011–2005 | Phase II 2016–2011 | Overall 2016–2005 |
| Benishangul | 20.28 | 23.47 | 2.94 | 3.19 | -20.53 | -17.34 |
| SNNPR | 15.52 | 10.34 | 7.12 | -5.18 | -3.22 | -8.4 |
| Gambela | 29.18 | 32.5 | 10.28 | 3.32 | -22.22 | -18.9 |
| Harari | 48.55 | 32.88 | 27.14 | -15.67 | -5.74 | -21.41 |
| Addis Ababa | 51.22 | 26.05 | 21.49 | -25.17 | -4.56 | -29.73 |
| Dire Dawa | 35.51 | 34.01 | 9.46 | -1.05 | -24.55 | -29.05 |

in the composition of women with wanted last pregnancy [β = 0.008962, 95% CI: 0.005355, 0.012569] contributes to the change of pre-lacteal feeding practice by 6.33%. Moreover, a decrease in the composition of women with primary education, working women, women who had three, and four and more ANC visits, with health facility delivery, and who initiated breastfeeding within one hour during the survey period showed a significant contribution to change of pre-lacteal feeding practice (**Table 4**).

After controlling the role of compositional changes, 79.69% of the decrease in pre-lacteal feeding practice was due to the difference in coefficients (the effects of characteristics) (**Table 3**). Factors including the perception of distance from the health facility, wealth status, occupation, ANC visit, and wanted last-child/pregnancy showed a significant effect on the observed change in pre-lacteal feeding practice. About 10.22% of the change in pre-lacteal feeding practice over the past decade was attributable due to the difference in the effect among women who did not perceive distance from the health facility as a big problem [β = 0.014483, 95% CI: 0.003535, 0.025430]. About 11.06% and 13.59% of the change in pre-lacteal feeding practice over the past decade was attributable due to the difference in the effect among women from middle [β = -0.015663, 95% CI: -0.025865, -0.005463] and rich households [β = -0.019250, 95% CI: -0.038553, -0.005230]. Compared with no ANC visit, a decrease in the effects of women with four or more ANC visits [β = -0.013735, 95% CI: -0.022427, -0.005044] contributes to the change in pre-lacteal feeding practice over the past decade by 9.70%. A decrease in the effects of women with wanted last-child/pregnancy [β = -0.023257, 95% CI: -0.037046, -0.009468], as compared to their counterparts, contributes to the change of pre-lacteal feeding practice over the past decade by 16.42% (**Table 4**).

## Spatial distribution of pre-lacteal feeding practice in Ethiopia, using EDHS 2016 data

**Spatial autocorrelation.** The spatial autocorrelation result revealed that pre-lacteal feeding practice in Ethiopia was non-random with Global Moran's I = 0.293 at p< 0.001 (**Fig 2**).

**Table 3. The overall decomposition analysis of the decrease in pre-lacteal feeding practice among reproductive-age women in Ethiopia, 2005 to 2016.**

| Prelacteal feeding | Coefficient | Percentage |
|---|---|---|
| E | -0.028772[-.041041, -.016503] * | 20.31 |
| C | -0.11288 [-.13371, -.092048] * | 79.69 |
| R | -0.14165[-.15739, -.12591] * | |

Note

* P-value<0.05, E: Endowment, C: Coefficient, R: Residual.

**Table 4. Decomposition of change in pre-lacteal feeding practice among reproductive-age women in Ethiopia, 2005 to 2016.**

| Characteristics | Difference due to characteristics (E) | | Difference due to coefficients (C) | |
|---|---|---|---|---|
| | Coefficient | Percent | Coefficient | Percent |
| Residence | | | | |
| Urban | 0.004617[.002943, .006291] * | -3.26 | 0.005931[-.0031512, .015014] | -4.19 |
| Rural | 0 | | 0 | |
| Distance from the health facility | | | | |
| Big problem | 0 | | 0 | |
| Not a big problem | 0.005564 [.002435, .008692] * | -3.93 | 0.014483[.003535, .025430] * | -10.22 |
| Age (years) | | | | |
| 15–24 | 0 | | 0 | |
| 25–34 | 0.000109[-.000591, .000810] | -0.08 | 0.003314[-.017903, .024531] | -2.34 |
| 35–49 | 0.000947[-.000728, .002623] | -0.67 | -0.013772[-.028716, .001173] | 9.72 |
| Educational level | | | | |
| No education | 0 | | 0 | |
| Primary | -0.006430[-.009668, -.003192] * | 4.54 | -0.006384[-.013862, .001095] | 4.51 |
| Secondary & above | -0.001083[-.002666,.000501] | 0.76 | -0.004352[-.010104, .001400] | 3.07 |
| Wealth status | | | | |
| Poor | 0 | | 0 | |
| Middle | 0.003016[.001606, .004425] * | -2.13 | -0.015663[-.025865, -.005463] * | 11.06 |
| Rich | 0.001239[.000101, .002378] | -0.88 | -0.019250[-.038553, -.005230] * | 13.59 |
| Occupation | | | | |
| Not working | 0 | | 0 | |
| Working | -0.003891[-.005836, -.001945] | 2.75 | -0.021776[-.032158, -.011393] * | 15.37 |
| Media exposure | | | | |
| No | 0 | | 0 | |
| Yes | -0.000013[-.000823, .000849] | -0.01 | -0.009382[-.025526, .006762] | 6.62 |
| Parity | | | | |
| Primiparous | 0 | | 0 | |
| Multiparous | -0.000002[-.000004, 0.000003] | 0.001 | -0.019034[-.039963, .001896] | 13.44 |
| Grand multiparous | 0.0005645[-.000476, .001605] | -0.4 | 0.002905[-.020378, .026189] | -2.05 |
| ANC visit | | | | |
| No | 0 | | 0 | |
| 01-Feb | -0.000318[-.001709, .001072] | 0.22 | -0.002408[-.006789, .001973] | 1.7 |
| 3 | -0.004930[-.008080, -.001779] * | 3.48 | -0.003173[-.007029, .000682] | 2.24 |
| 4 and above | -0.014985[-.020747, -.009223] * | 10.58 | -0.013735[-.022427, -.005044] * | 9.7 |
| Place of delivery | | | | |
| Home | 0 | | 0 | |
| Health facility | -0.011096[-.018689, -.003502] * | 7.83 | -0.003504[-.006518, .016576] | 2.47 |
| Delivery by CS | | | | |
| No | 0 | | 0 | |
| Yes | 0.000628[-.000057, .001313] | -0.44 | -0.000057[-.001984,.001869] | 0.04 |
| Size of the child at birth | | | | |
| large | -0.000241[-.000418, -.000064] * | 0.17 | 0.005029[-.000015, .022246] | -3.55 |
| Average/normal | 0 | | 0 | |
| small | 0.000197[.000060, .000334] * | -0.14 | 0.011115[-.000015, .022246] | -7.85 |
| Sex of the child | | | | |
| Male | 0 | | 0 | |
| Female | -0.000131[-.000368,.000105] | 0.09 | -0.010325[-.026199, .005549] | 7.29 |

(*Continued*)

**Table 4.** (Continued)

| Characteristics | Difference due to characteristics (E) | | Difference due to coefficients (C) | |
|---|---|---|---|---|
| | Coefficient | Percent | Coefficient | Percent |
| Timing of BF initiation | | | | |
| Within one hour | -0.011522[-.013519, -.009525] * | 8.134 | -0.012237[-.037767, .013294] | 8.64 |
| After one hour | 0 | | 0 | |
| Wanted of the child | | | | |
| Yes | 0.008962[.005355, .012569] * | -6.33 | -0.023257[-.037046, -.009468] * | 16.42 |
| No | 0 | | 0 | |

Note

\* = p value < 0.05.

**Kriging interpolation.** The kriging interpolation result revealed that regions such as Benishangul, Tigray, most parts of Amhara, the western part of Gambela, and eastern parts of SNNPR had predicted lower rates of pre-lacteal feeding practice. However, the Somalia region and the Afar region had higher predicted rates of pre-lacteal feeding practice (**Fig 3**).

**Hotspot and cold spot analysis.** Fig 4 revealed the hot spot analysis of pre-lacteal feeding practice in Ethiopia. The red color indicates regions with significant hotspot areas (areas with high rates of pre-lacteal feeding practice), which were found in the Afar and Somalia regions. The blue color indicates areas/regions with significantly lower rates of pre-lacteal feeding practice (cold spot areas), which were found in Oromia, Benishangul, Tigray, and in central parts of the Amhara region (**Fig 4**).

**SaTScan analysis (Bernoulli based model).** One hundred five significant clusters (48 primary and 57 secondary clusters) were identified in the SaTScan analysis. The primary clusters spatial window was located in the Somalia region, which was centered at 6.641319 N, 44.092837 E geographic location with 360.78 km radius, and LLR of 123.18 at p < 0.001. The relative risk (RR) of the primary clusters spatial window was 3.81 and this revealed that women within the spatial window had 3.81 times higher risk of pre-lacteal feeding practice than women outside the window. The secondary clusters scanning window was located in the Afar region, which was centered at 12.401068 N, 42.163134 E geographic location with 305.05 km radius, and LLR of 58.58 at p-value <0.001. The RR value was 2.67 and this showed that women within the spatial window had 2.67 times higher risk of pre-lacteal feeding practice than women outside the window (**Table 5**, **Fig 5**).

## Discussion

This study aimed to assess the spatial distribution and determinants of the change in pre-lacteal feeding practice over time in Ethiopia.

About one-fifth (20.31%) of the overall change in pre-lacteal feeding practice in Ethiopia was due to difference in characteristics. The reason for this was associated with the significant change in the structural composition of women who participated in the surveys.

An increase in the proportion of urban women and women who did not perceive distance from the health facility as a big problem in the sample made a significant contribution to the change of pre-lacteal feeding practice. This might indicate urban women are mostly exposed to information regarding optimal breastfeeding practices. In addition, this could mean that women living in urban areas have a greater awareness and use of maternal health services [32–34]. Maternal health services such as delivery service, however, are not readily available for

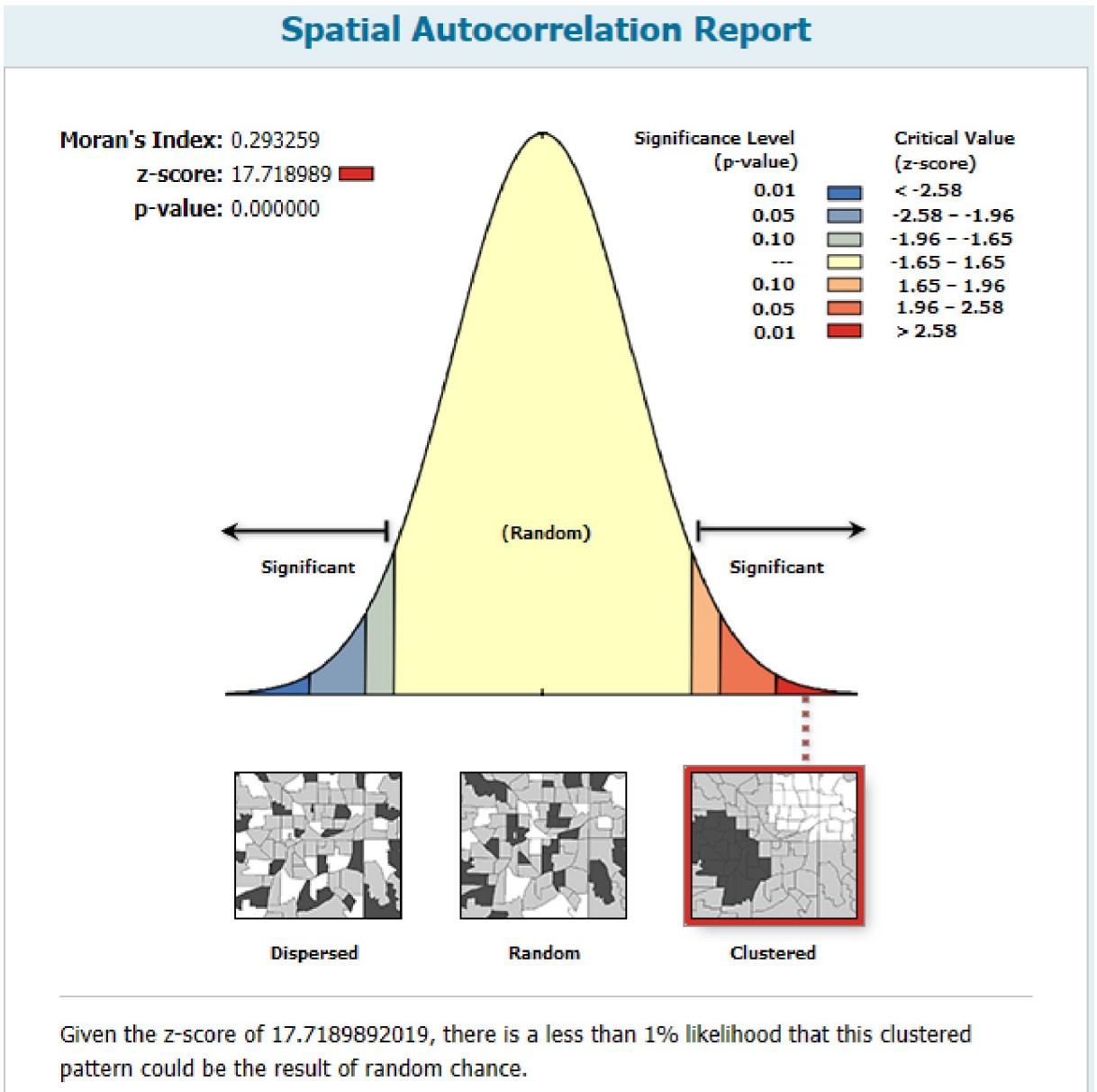

**Fig 2. Spatial autocorrelation result of pre-lacteal feeding practice in Ethiopia, 2016.**

mothers from remote areas, which in turn decrease awareness about optimal breastfeeding and raise pre-lacteal feeding practice [35].

A decrease in the composition of women who had three and four and more ANC visits over the survey period contributes to the change of pre-lacteal feeding practice. Besides, decreasing the composition of women who gave birth in the health facility over time contributes to the change in pre-lacteal feeding practice. This result might indicate women with ANC visits and delivery at the health facility may have a chance to obtain information on appropriate breastfeeding practices and avoid giving of pre-lacteal foods to the newborn [32,34].

Regarding the timing of the initiation of breastfeeding, a decrease in the composition of women who initiated breastfeeding within one hour contributes to the change in pre-lacteal

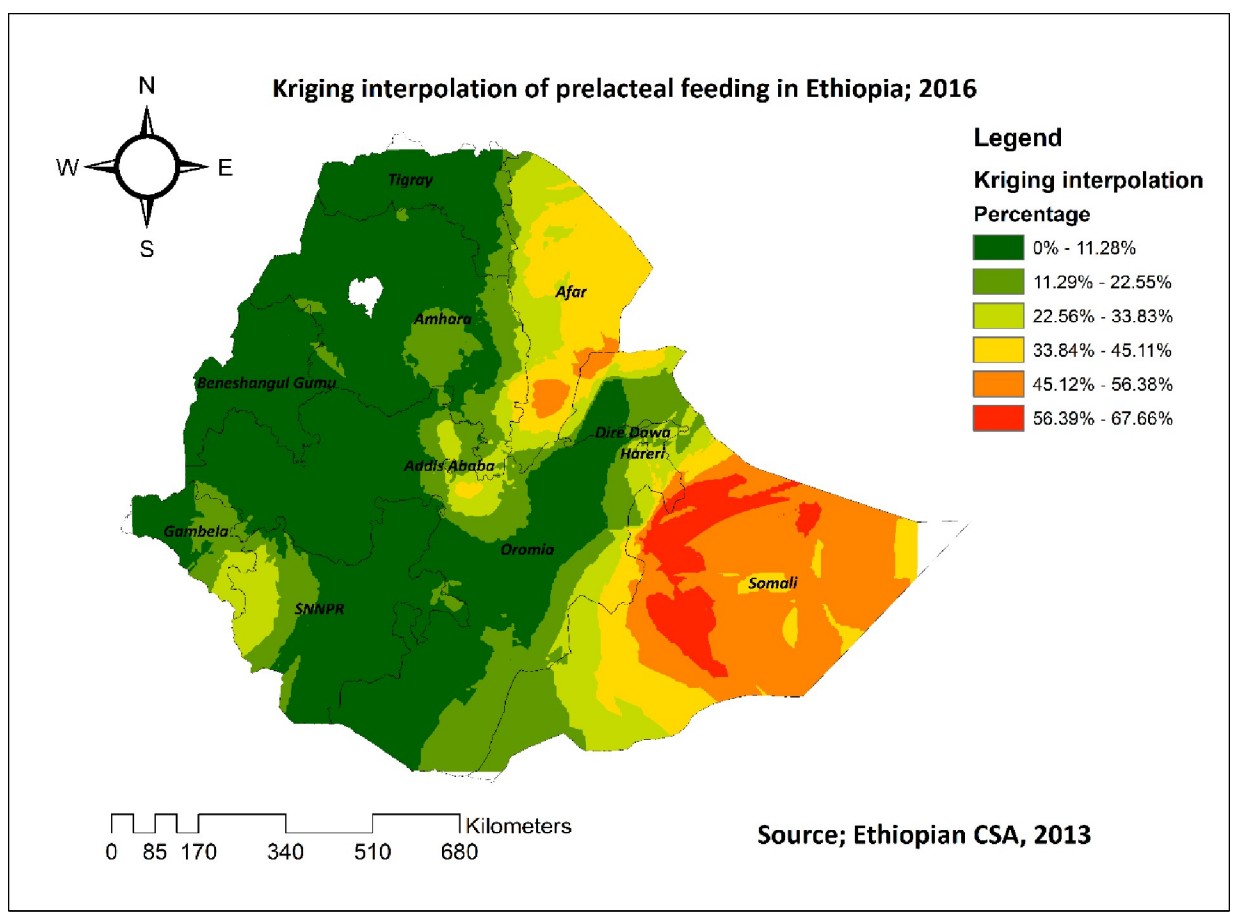

**Fig 3. Kriging interpolation of pre-lacteal feeding practice in Ethiopia, 2016.**

feeding practice over the survey period. This may indicate that women with early breastfeeding initiation have no room for additional feedings, such as pre-lacteal foods for the newborn [15,36,37].

The study at hand also revealed that a decrease in the composition of women with wanted last pregnancy over time contributes to the change in pre-lacteal feeding practice. This can reflect unintended pregnancy, which may result in low use of maternal health services, can contribute to suboptimal breastfeeding such as pre-lacteal feeding [38].

Moreover, a decrease in the composition of women's attainment of primary education and a decrease in the proportion of working women during the survey period showed a significant contribution to the increment of the differential of pre-lacteal feeding practice. Also, an increasing proportion of women from households with a middle wealth index over the survey period contributes to a significant decrease in pre-lacteal feeding practice.

In this study, about four-fifth (79.69%) of the decrease in pre-lacteal feeding practice over the past decade was due to differences in the effects of characteristics (coefficients). About 10.22% of the decrease in pre-lacteal feeding practice over the past decade was attributable due to the difference in the effect of not perceiving distance from the health facility as a big problem. This is supported by a study, which reports that an increased distance from the health facility is associated with increasing pre-lacteal feeding practice [25]. This may be because women from remote areas are unable to access maternal health services and are unable to

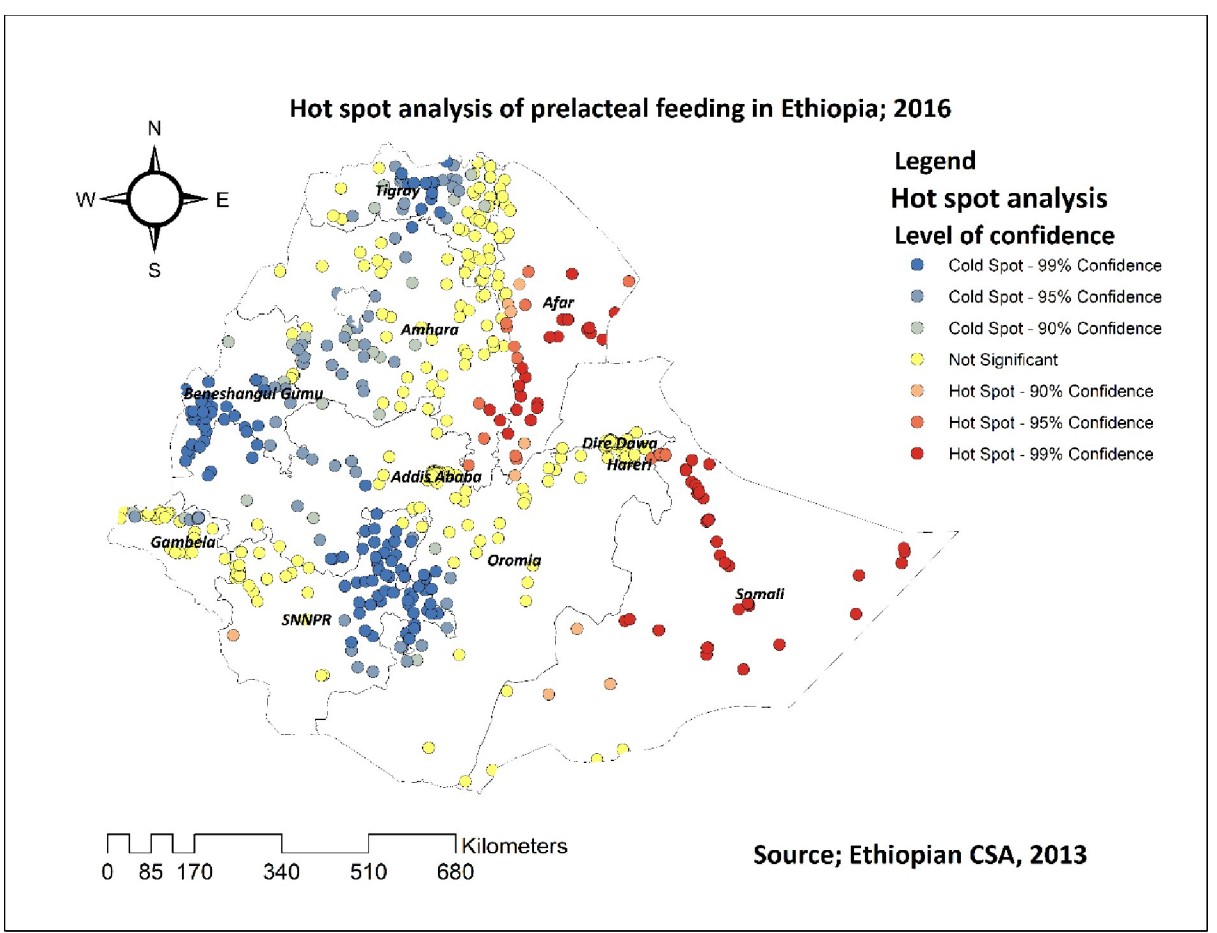

**Fig 4. Hot spot and cold spot analysis of pre-lacteal feeding practice in Ethiopia, 2016.**

access schooling. This might in turn result in lower awareness about optimal breastfeeding and increased the practice of pre-lacteal feeding.

The study at hand also revealed that about 11.06% and 13.59% of the change in pre-lacteal feeding practice over the past decade was due to changes in pre-lacteal feeding behavior of women from middle and rich households, respectively. Other studies in Ethiopia also revealed that women with improved socioeconomic status had a lower chance of practicing pre-lacteal feeding [29,39]. This could be due to mothers with improved socioeconomic status are mostly educated and can easily access maternal health services, such as getting advice on optimal breastfeeding practices.

Compared with no ANC visit, the effects of being having four or more ANC visits were a significant predictor for the change in pre-lacteal feeding practice over the past decade. This is in line with different studies [21,26], which revealed that having an ANC visit is associated with lower risks of pre-lacteal feeding practice. This might be because having an ANC visit might

**Table 5. Significant clusters of areas with high pre-lacteal feeding practice in Ethiopia, 2016.**

| Number of significant clusters (Total = 105) | Coordinates/radius | population | case | RR | LLR | P-value |
|---|---|---|---|---|---|---|
| 48 (primary) | (6.641319 N, 44.092837 E) / 360.78 km | 394 | 186 | 3.81 | 123.18 | <0.001 |
| 57 (secondary) | (12.401068 N, 42.163134 E) / 305.05 km | 423 | 152 | 2.67 | 58.58 | <0.001 |

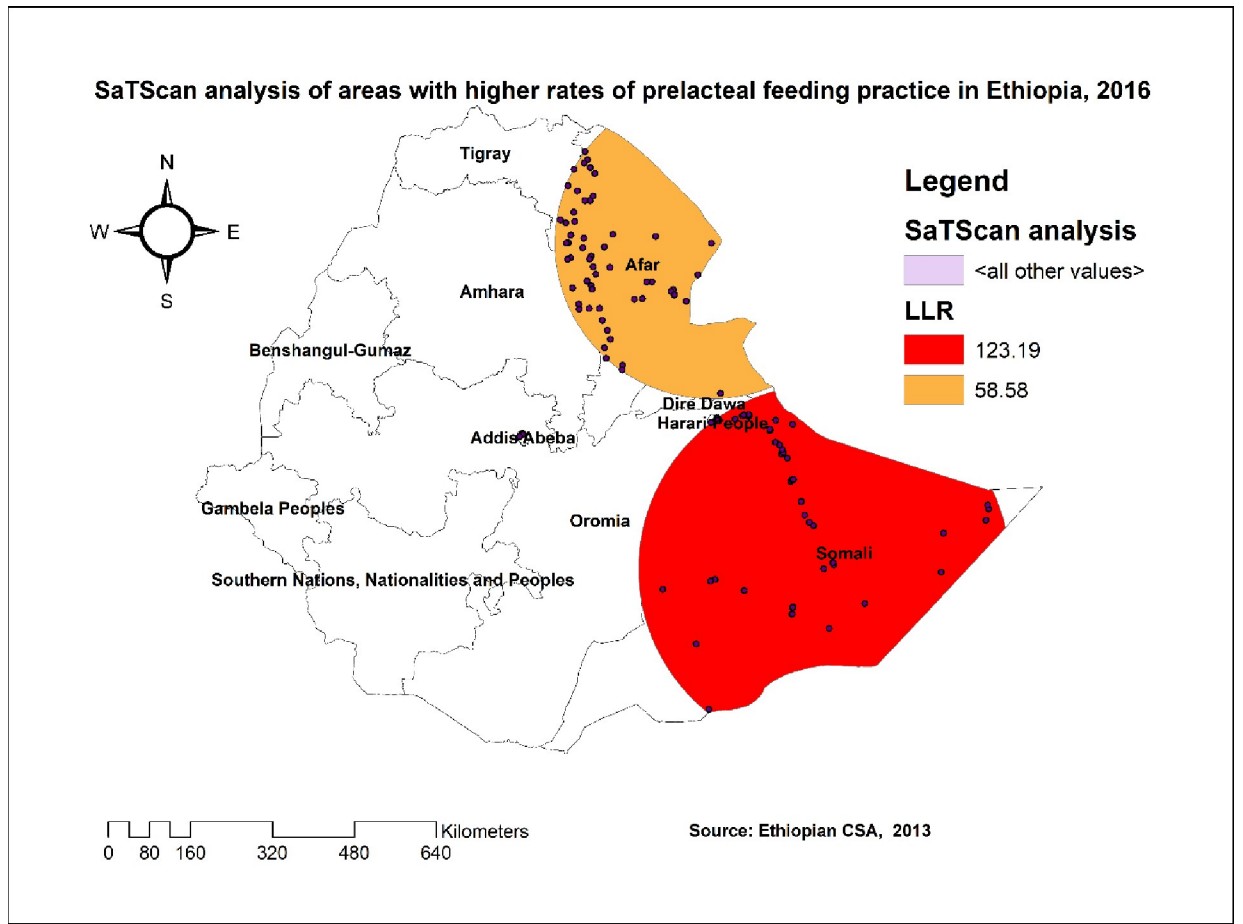

**Fig 5. SaTScan analysis of pre-lacteal feeding practice in Ethiopia, 2016.**

expose a woman to information regarding newborn health and breastfeeding practices. Moreover, as compared to being unwanted last last-child/pregnancy, a decrease in the effect of being wanted last-child/pregnancy was associated with the change of pre-lacteal feeding practice over the past decade. This may be because women with a desired pregnancy pay greater attention to their pregnancy and utilize maternal health services for the newborn's health, which could result in the mother having exposure to pre-lacteal feeding and its negative impact on the newborn.

In this study, cesarean delivery was not associated with the change in pre-lacteal feeding practice (both in the endowment and coefficient parts). However, multiple studies have shown that delivery by cesarean section has a strong association with pre-lacteal feeding practice [18,40]. The discrepancy may be because this study was a decomposition analysis in which the trend was analyzed (factors for the change in pre-lacteal feeding practice over time was assessed); whereas the other studies were carried out using a binary logistic regression (did not assess the trend). The author does, however, suggest further studies in this regard.

The spatial analysis revealed that the spatial distribution of pre-lacteal feeding practice in 2016 was non-random in Ethiopia. The SaTScan analysis result revealed that the primary clusters spatial window was found in the Somali region and the secondary clusters spatial window was found in the Afar region. The hot spot analysis result also revealed that these regions had higher rates of pre-lacteal feeding practice. This finding, regional variations of pre-lacteal feeding practice, was supported by different studies conducted in Ethiopia [21], and Nepal [13,41].

This might be because these regions are found in border areas of Ethiopia in which maternal health services are not easily accessible.

This study presented important findings to minimize pre-lacteal feeding practice in Ethiopia since it identified areas with higher rates of pre-lacteal feeding practice using spatial analysis. Besides, the study identified the factors that contributed to the change in pre-lacteal feeding practice over time using decomposition analysis. Nevertheless, this study was not without limitations. Due to a lack of studies on pre-lacteal feeding practice, using decomposition analysis, we were forced to consider studies conducted on pre-lacteal feeding practice in general while discussing our findings. We did not also consider important variables such as maternal beliefs and maternal knowledge towards breastfeeding since these factors were not found in the survey.

## Conclusion

Pre-lacteal feeding practice was significantly decreased over the 10-year period. The decomposition analysis revealed that about one-fifth (20.31%) and four-fifth (79.69%) of the overall change in pre-lacteal feeding practice in Ethiopia was due to the difference in characteristics and coefficients, respectively. Therefore, program interventions considering women with no ANC visit, women with poor socioeconomic status, women with an unintended pregnancy, and women from remote areas especially at border areas such as Somali and Afar could decrease pre-lacteal feeding practice in Ethiopia.

## Acknowledgments

We would like to acknowledge the MEASURE DHS program, which helps us to access and use the data sets.

## Author Contributions

**Conceptualization:** Achamyeleh Birhanu Teshale, Misganaw Gebrie Worku, Getayeneh Antehunegn Tesema.

**Data curation:** Achamyeleh Birhanu Teshale, Misganaw Gebrie Worku, Getayeneh Antehunegn Tesema.

**Formal analysis:** Achamyeleh Birhanu Teshale, Misganaw Gebrie Worku, Getayeneh Antehunegn Tesema.

**Investigation:** Achamyeleh Birhanu Teshale, Misganaw Gebrie Worku, Getayeneh Antehunegn Tesema.

**Methodology:** Achamyeleh Birhanu Teshale, Misganaw Gebrie Worku, Getayeneh Antehunegn Tesema.

**Resources:** Achamyeleh Birhanu Teshale, Misganaw Gebrie Worku, Getayeneh Antehunegn Tesema.

**Software:** Achamyeleh Birhanu Teshale, Misganaw Gebrie Worku, Getayeneh Antehunegn Tesema.

**Validation:** Achamyeleh Birhanu Teshale, Misganaw Gebrie Worku, Getayeneh Antehunegn Tesema.

**Visualization:** Achamyeleh Birhanu Teshale, Misganaw Gebrie Worku, Getayeneh Antehunegn Tesema.

**Writing – original draft:** Achamyeleh Birhanu Teshale, Misganaw Gebrie Worku, Getayeneh Antehunegn Tesema.

**Writing – review & editing:** Achamyeleh Birhanu Teshale, Misganaw Gebrie Worku, Getayeneh Antehunegn Tesema.

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
