## [Decision Letter · Decision Letter 0]

16 Nov 2020

PONE-D-20-27150

Spatial distribution and determinants of the change in prelacteal feeding practice in Ethiopia; a spatial and multivariate decomposition analysis

PLOS ONE

Dear Dr. Teshale,

Thank you for submitting your manuscript to PLOS ONE. After careful consideration, we feel that it has merit but does not fully meet PLOS ONE’s publication criteria as it currently stands. Therefore, we invite you to submit a revised version of the manuscript that addresses the points raised during the review process.

We look forward to receiving your revised manuscript.

Kind regards,

Hannah Tappis, DrPH, MPH

Academic Editor

PLOS ONE

3. In statistical methods, please clarify whether you corrected for multiple comparisons-

4. As part of your revision, please complete and submit a copy of the STROBE checklist, a document that aims to improve reporting and reproducibility of observational studies for purposes of post-publication data analysis and reproducibility: (http://www.strobe-statement.org). Please include your completed checklist as a Supporting Information file. Note that if your paper is accepted for publication, this checklist will be published as part of your article.

5. We note that Figures 4, 5, 6 and 7 in your submission contain map images which may be copyrighted. All PLOS content is published under the Creative Commons Attribution License (CC BY 4.0), which means that the manuscript, images, and Supporting Information files will be freely available online, and any third party is permitted to access, download, copy, distribute, and use these materials in any way, even commercially, with proper attribution. For these reasons, we cannot publish previously copyrighted maps or satellite images created using proprietary data, such as Google software (Google Maps, Street View, and Earth). For more information, see our copyright guidelines: http://journals.plos.org/plosone/s/licenses-and-copyright.

5.1.    You may seek permission from the original copyright holder of Figures 4, 5, 6 and 7 to publish the content specifically under the CC BY 4.0 license. 

5.2.    If you are unable to obtain permission from the original copyright holder to publish these figures under the CC BY 4.0 license or if the copyright holder’s requirements are incompatible with the CC BY 4.0 license, please either i) remove the figure or ii) supply a replacement figure that complies with the CC BY 4.0 license. Please check copyright information on all replacement figures and update the figure caption with source information. If applicable, please specify in the figure caption text when a figure is similar but not identical to the original image and is therefore for illustrative purposes only.

Reviewers' comments:

Reviewer's Responses to Questions

**Comments to the Author**

1. Is the manuscript technically sound, and do the data support the conclusions?

Reviewer #1: Yes

Reviewer #2: Yes

2. Has the statistical analysis been performed appropriately and rigorously? 

Reviewer #1: Yes

Reviewer #2: Yes

3. Have the authors made all data underlying the findings in their manuscript fully available?

Reviewer #1: Yes

Reviewer #2: Yes

4. Is the manuscript presented in an intelligible fashion and written in standard English?

Reviewer #1: Yes

Reviewer #2: Yes

5. Review Comments to the Author

Reviewer #1: Date 4 November 2019

Dear Editor of BMC Public Health

Thank you so much for giving me the opportunity to review this important paper.

This paper uses pooled data of Ethiopia demography and health surveys with the aims of determining the determinants of pre-lacteal feeding practices in Ethiopia. In addition, the paper investigates the factors which explains the prelacteal practices and depicts the trends of prelacteal feeding practices behavior in Ethiopia. Overall, the paper is interesting as it deals with the prelacteal feeding practices which can explain the current mal-practices and shows future program focus areas. Moreover, Ethiopia is striving to further reduce neonatal and infant mortality rates. The author gives a clear background on the statement of problems. And clearly describe the source of data which is an ideal sources to determine the national estimates. In the result and discussions section the author presents the point estimates, trends and factors influencing behaviors of parents or care givers.

Some of the strengths of this manuscript are:

• The paper addressed public health important issue.

• Uses data from DHS, which are nationally representative cross-sectional surveys

• Use proper scientific writings steps

• Uses proper statistical analysis

• Uses advanced level of English language writing

• The results and discussions answer the objective

• The conclusion is in line with the objectives and results presented.

Best regards,

Reviewer #2: Introduction:

The starting paragraphs mostly deal with exclusive breastfeeding and delayed breastfeeding. However, this paper deals with prelacteal feeding practices. I would suggest starting introduction with prelacteal feeding and then connect prelacteal feeding with delayed breastfeeding and lack of exclusive breastfeeding. The readers would be interested to get to know about preleacteal to start of introduction.

Methods:

Method section is well explained. I am just curious that the authors used a number of independent variables i.e. region, residence, perception of distance from the health facility, age, educational level, wealth index, occupation, media exposure, parity, ANC visit, place of delivery, delivery by cesarean section, size of the child at birth, and timing of initiation of breastfeeding and authors ONLY explained two variables in operational definition section.

Result

The authors presented the results in very details. It is a bit difficult to read and digest the results of all variables. It is also a good idea to just explain your major results and the reader can get the idea of detailed results from the table, and figure and be focused on your major findings. Decrement word is repeatedly used that may be changed to another appropriate word.

Discussion:

Discussion section can be improved; for example line 303-306 early initiation of breastfeeding is considered the window of opportunity to decrease the prelacteal feeding. It is important to discuss more and cite some relevant studies to improve this practice. The authors empahsized a lot in the introduction section and they did not put weigh in the discussion section.

Furthermore, different studies report that Delivery by CS is the major determinant of initiation of prelacteal feeding practices in neonate. This study does not report this. It may be important to explain why?

6. PLOS authors have the option to publish the peer review history of their article (what does this mean?). If published, this will include your full peer review and any attached files.

Reviewer #1: **Yes: **Mesele Damte Argaw, PhD

Reviewer #2: **Yes: **Muhammad Asim

---

## [Author Response · Author response to Decision Letter 0]

4 Dec 2020

Date: December 4, 2020

Point by point response to editor and reviewer comments 

Title: Spatial distribution and determinants of the change in pre-lacteal feeding practice in Ethiopia; a spatial and multivariate decomposition analysis

Manuscript number: PONE-D-20-27150

Dear editor and Reviewers: We really thank you for your valuable comments for the betterment of our manuscript. Your concerns and questions as well as suggestions are addressed in the revised manuscript. 

Response to Editorial comment/journal requirement 

Author’s response: The author confirm that the revised manuscript meets PLOS ONE's style.

2. We suggest you thoroughly copyedit your manuscript for language usage, spelling, and grammar

Author’s response: We extensively edited the manuscript after consulting our colleagues and language experts who had MA degree in TEAFL (teaching English as foreign language) and who had many years’ experience in the area of literature at University of Gondar. A copy of our manuscript showing the changes is indicated by using track changes (See supporting information file). 

3. In statistical methods, please clarify whether you corrected for multiple comparisons-

Author’s response: Dear editor we consider it in the revised manuscript. In this study, the trends (overall and per each categories of independent variables) were described in the descriptive analysis. As you know in the decomposition analysis, there are two parts, the endowment and the coefficient parts, and we interpreted and discussed the results separately. Moreover, the spatial analysis was conducted using the recent EDHS (EDHS 2016) data. 

4. As part of your revision, please complete and submit a copy of the STROBE checklist, a document that aims to improve reporting and reproducibility of observational studies for purposes of post-publication data analysis and reproducibility: (http://www.strobe-statement.org). Please include your completed checklist as a Supporting Information file. Note that if your paper is accepted for publication, this checklist will be published as part of your article.

Author’s response: Thank you. We incorporated the STROBE checklist as supporting information (see supporting information).

5. We note that Figures 4, 5, 6 and 7 in your submission contain map images which may be copyrighted. All PLOS content is published under the Creative Commons Attribution License (CC BY 4.0), which means that the manuscript, images, and Supporting Information files will be freely available online, and any third party is permitted to access, download, copy, distribute, and use these materials in any way, even commercially, with proper attribution. For these reasons, we cannot publish previously copyrighted maps or satellite images created using proprietary data, such as Google software (Google Maps, Street View, and Earth).

Author’s response: Thank you for the comment. These figures are not copyrighted from other sources rather they are our findings using Arc-GIS version 10.3 and SaTScan version 9.6 statistical softwares. The shape file of Ethiopia was found in the website https://africaopendata.org/dataset/ethiopia-shapefiles, and then we generate the figures using the GPs (latitude and longitude) data and the outcome variable using ArcGIS version 10.3 and SaTScan version 9.6 statistical softwares. So all the figures are not copyrighted form other source rather we generate using the software. 

Response to reviewers

Reviewer #1: Dear reviewer thank you in advance for reviewing our paper.

Reviewer #2: 

1. Introduction

The starting paragraphs mostly deal with exclusive breastfeeding and delayed breastfeeding. However, this paper deals with pre-lacteal feeding practices. I would suggest starting introduction with pre-lacteal feeding and then connect pre-lacteal feeding with delayed breastfeeding and lack of exclusive breastfeeding. The readers would be interested to get to know about pre-lacteal to start of introduction.

Author’s response: Dear reviewer thank you for this important concern you raised. We consider you comment and we bother about pre-lacteal feeding in the introduction/background section. We also indicate the relation of pre-lacteal feeding with delayed breastfeeding and lack of exclusive breastfeeding.

2. Methods

Method section is well explained. I am just curious that the authors used a number of independent variables i.e. region, residence, perception of distance from the health facility, age, educational level, wealth index, occupation, media exposure, parity, ANC visit, place of delivery, delivery by cesarean section, size of the child at birth, and timing of initiation of breastfeeding and authors ONLY explained two variables in operational definition section.

Author’s response: Thank you for the comment. We operationalized media exposure and size of the child at birth to make them measurable and to indicate how these variables were measured to the international readers. However, the rest of the variables are straightforward and there categories are found in the result section (in the tables). Dear reviewer if you are not convinced we are open to consider your comment again. 

3. Result

The authors presented the results in very details. It is a bit difficult to read and digest the results of all variables. It is also a good idea to just explain your major results and the reader can get the idea of detailed results from the table, and figure and be focused on your major findings. Decrement word is repeatedly used that may be changed to another appropriate word.

Author’s response: Thank you for your constructive comment. We put the major findings by avoiding extra and detailed results in the result section of the revised manuscript. In addition, we consider other words for some repeatedly stated words such as “Decrement”. 

4. Discussion

Discussion section can be improved; for example line 303-306 early initiation of breastfeeding is considered the window of opportunity to decrease the pre-lacteal feeding. It is important to discuss more and cite some relevant studies to improve this practice. The authors emphasized a lot in the introduction section and they did not put weigh in the discussion section.

Furthermore, different studies report that Delivery by CS is the major determinant of initiation of pre-lacteal feeding practices in neonate. This study does not report this. It may be important to explain why?

Author’s response: Dear reviewer thank you for the important concern you raised. We consider your comment in the revised manuscript. In addition, Cesarean delivery was not associated with the change in pre-lacteal feeding practice in this study (unlike that of the previous studies) and we put a one paragraph statement reveling the discrepancy, the possible reason and further recommendation on this regard (see the discussion section paragraph 10 line 335-341 ).

---

## [Editor Report · Decision Letter 1]

10 Dec 2020

PONE-D-20-27150R1

Spatial distribution and determinants of the change in pre-lacteal feeding practice over time in Ethiopia: a spatial and multivariate decomposition analysis

PLOS ONE

Dear Dr. Teshale,

Thank you for submitting your manuscript to PLOS ONE. After careful review, we feel that while initial reviewer feedback was addressed, additional minor revisions are needed to consider this manuscript for publication. Therefore, we invite you to submit a revised version of the manuscript that addresses the points raised during the review process.

We look forward to receiving your revised manuscript.

Kind regards,

Hannah Tappis, DrPH, MPH

Academic Editor

PLOS ONE

Additional Editor Comments (if provided):

* In the Methods section, where operational definitions are presented, it would be prudent to add a sentence similar to that included in the response to reviewer comments, noting that "Other independent variable definitions are self-explanatory" and also citing standard DHS survey modules for further reference.

* Figures 1 and 3 are superfluous (though data on regional distributions is relevant and important). Please consider omitting these figures and integrating regional distribution data as a row in Tables 1 and 2 respectively.

---

## [Author Response · Author response to Decision Letter 1]

11 Dec 2020

Date: December 11, 2020

Response to editor comment 

Title: Spatial distribution and determinants of the change in pre-lacteal feeding practice over time in Ethiopia: a spatial and multivariate decomposition analysis

Manuscript number: PONE-D-20-27150

Dear editor: We really thank you for your valuable comments for the betterment of our manuscript. Your concerns and questions as well as suggestions are addressed in the revised manuscript.

Point by point response to editor comment 

1. In the Methods section, where operational definitions are presented, it would be prudent to add a sentence similar to that included in the response to reviewer comments, noting "Other independent variable definitions are self-explanatory" and also citing standard DHS survey modules for further reference.

Author’s response: We added the sentence and we put the survey module as a reference (See line 120 and 121 of the revised manuscript). 

2. Figures 1 and 3 are superfluous (though data on regional distributions is relevant and important). Please consider omitting these figures and integrating regional distribution data as a row in Tables 1 and 2 respectively.

Author’s response: Thank you. We consider the comment (see the revised manuscript).

---

## [Editor Report · Decision Letter 2]

14 Dec 2020

Spatial distribution and determinants of the change in pre-lacteal feeding practice over time in Ethiopia: a spatial and multivariate decomposition analysis

PONE-D-20-27150R2

Dear Dr. Teshale,

We’re pleased to inform you that your manuscript has been judged scientifically suitable for publication and will be formally accepted for publication once it meets all outstanding technical requirements.

Kind regards,

Hannah Tappis, DrPH, MPH

Academic Editor

PLOS ONE
---

## [Editor Report · Acceptance letter]

4 Jan 2021

PONE-D-20-27150R2 

Spatial distribution and determinants of the change in pre-lacteal feeding practice over time in Ethiopia: a spatial and multivariate decomposition analysis 

Dear Dr. Teshale:

I'm pleased to inform you that your manuscript has been deemed suitable for publication in PLOS ONE. Congratulations! Your manuscript is now with our production department. 

Kind regards, 

on behalf of

Dr. Hannah Tappis 

Academic Editor

PLOS ONE